# Application of Repetitive Transcranial Magnetic Stimulation in Neuropathic Pain: A Narrative Review

**DOI:** 10.3390/life13020258

**Published:** 2023-01-17

**Authors:** Yuan-Yuan Tsai, Wei-Ting Wu, Der-Sheng Han, Kamal Mezian, Vincenzo Ricci, Levent Özçakar, Po-Cheng Hsu, Ke-Vin Chang

**Affiliations:** 1Department of Physical Medicine and Rehabilitation, National Taiwan University Hospital, Bei-Hu Branch, Taipei 10845, Taiwan; 2Department of Physical Medicine and Rehabilitation, National Taiwan University College of Medicine, Taipei 10048, Taiwan; 3Department of Rehabilitation Medicine, First Faculty of Medicine, General University Hospital in Prague, Charles University, 12800 Prague, Czech Republic; 4Physical and Rehabilitation Medicine Unit, Luigi Sacco University Hospital, ASST Fatebenefratelli-Sacco, 20157 Milan, Italy; 5Department of Physical and Rehabilitation Medicine, Hacettepe University Medical School, Ankara 06100, Turkey; 6Department of Physical Medicine and Rehabilitation, West Garden Hospital, Taipei 10864, Taiwan; 7Center for Regional Anesthesia and Pain Medicine, Wang-Fang Hospital, Taipei Medical University, Taipei 11600, Taiwan

**Keywords:** rTMS, neuropathic pain, analgesia, neuromodulation, rehabilitation

## Abstract

Neuropathic pain, affecting 6.9–10% of the general population, has a negative impact on patients’ quality of life and potentially leads to functional impairment and disability. Repetitive transcranial magnetic stimulation (rTMS)—a safe, indirect and non-invasive technique—has been increasingly applied for treating neuropathic pain. The mechanism underlying rTMS is not yet well understood, and the analgesic effects of rTMS have been inconsistent with respect to different settings/parameters, causing insufficient evidence to determine its efficacy in patients with neuropathic pain. This narrative review aimed to provide an up-to-date overview of rTMS for treating neuropathic pain as well as to summarize the treatment protocols and related adverse effects from existing clinical trials. Current evidence supports the use of 10 Hz HF-rTMS of the primary motor cortex to reduce neuropathic pain, especially in patients with spinal cord injury, diabetic neuropathy and post-herpetic neuralgia. However, the lack of standardized protocols impedes the universal use of rTMS for neuropathic pain. rTMS was hypothesized to achieve analgesic effects by upregulating the pain threshold, inhibiting pain impulse, modulating the brain cortex, altering imbalanced functional connectivity, regulating neurotrophin and increasing endogenous opioid and anti-inflammatory cytokines. Further studies are warranted to explore the differences in the parameters/settings of rTMS for treating neuropathic pain due to different disease types.

## 1. Introduction

Neuropathic pain has been defined as “pain caused by lesions or diseases of the somatosensory nervous system” by International Association for the Study of Pain [1]. It has a negative impact on patients’ quality of life and potentially leads to functional impairment and disability. Neuropathic pain affects 6.9–10% of the general population [2], 11–57% of people with stroke [3], 53% of people with spinal cord injury (SCI) [4], 26% of people with type 2 diabetes [2] and 21% of people with post-herpes zoster infection [2].

The comprehensive management of neuropathic pain relies on the multidisciplinary approach with a biopsychosocial framework. Conventional pain management includes pharmacotherapy (such as acetaminophen, non-steroidal anti-inflammatory drugs, opioid and neuropathic agents), physical therapy (such as thermotherapy, electrotherapy, hydrotherapy and therapeutic exercises), psychological therapies (such as cognitive behavioral therapy and relaxation techniques), alternative therapies (such as acupuncture, massage and meditation) and injections (such as anesthetics, corticosteroid [5] and dextrose [6]).

Neurostimulation—recently applied in treating neuropathic pain—includes peripheral nerve stimulation [7], dorsal root ganglion stimulation [8], spinal cord stimulation and motor cortex stimulation (MCS) [9], transcranial direct current stimulation (tDCS) [10] and repetitive transcranial magnetic stimulation (rTMS) [11]. Among them, rTMS is a safe, indirect and non-invasive technique that produces a transient magnetic field to induce electrical current for modulating excitability in the brain cortex [12]. It is reported that low-frequency (≦1 Hz) repetitive transcranial magnetic stimulation (LF-rTMS) elicits inhibitory effects [13] and high-frequency (≧5 Hz) repetitive transcranial magnetic stimulation (HF-rTMS) incurs excitatory effects [14]. Based on the aforementioned basic principles, rTMS has been widely used in neuroscientific research and various clinical conditions encompassing depression, anxiety, psychiatric diseases, post-stroke complications (motor deficits, aphasia and dysphagia), movement disorders, consciousness disorders, cognitive impairments and pain [15].

rTMS has been applied for alleviating different pain scenarios, such as neuropathic pain, fibromyalgia and myofascial pain syndrome. An antecedent meta-analysis showed a positive analgesic effect of rTMS for patients with neuropathic pain in comparison with sham treatment [16]. However, the settings of rTMS (such as the intervention site, stimulation frequency, intensity and session) were heterogeneous, and the best protocol for relief of neuropathic pain seems warranted. Therefore, we provide an up-to-date overview of rTMS in treating neuropathic pain and related adverse effects. We summarize the respective treatment protocols for various neuropathic pain types caused by a single disease or lesion from existing clinical trials in order to find out identical parameters to guide clinicians in clinical practice.

## 2. Literature Search

Studies in the present review were identified by searching two electronic databases, PubMed and Embase, published from the earliest record to 31 October 2022. The key terms included “rTMS OR repetitive transcranial magnetic stimulation” and “neuropathic pain”, with additional restriction to human clinical trials or randomized controlled trials. No language restrictions were applied. Studied were included according to the criteria of (1) clinical trials or randomized controlled trials and (2) specific neuropathic pain caused by a single disease or lesion. Non-relevant studies were excluded by title or abstract (Figure 1). Furthermore, the reference lists of eligible articles were manually searched for additional relevant studies. A relatively small number of studies were enrolled, and selection bias may exist. Indications for rTMS in neuropathic pain are listed below in Table 1.

## 3. Therapeutic Application of rTMS in Neuropathic Pain

### 3.1. Central Post-Stroke Pain

Central post-stroke pain is difficult to treat. There are multifactorial etiologies contributing to the development of central post-stroke pain. The parameters of rTMS varied across different studies without reaching a consensus. All these factors complicate the treatment course. Not all patients respond to rTMS targeted on the primary motor cortex [17].

de Oliveira et al. found that applying 10 daily sessions of 10 Hz HF-rTMS to the premotor cortex/dorsolateral prefrontal cortex (DLPFC) at 120% of the resting motor threshold (RMT) in patients with central post-stroke pain did not cause significant analgesic effects compared to sham treatment at 1, 2 and 4 weeks after the last treatment session. Interestingly, variations in the daily visual analog scale (VAS) of pain indicated significant but non-sustained pain reduction right after each stimulation session in 5 out of 10 stimulation days [18]. Ojala et al. compared the analgesic effects of rTMS applied to the primary motor cortex with those of rTMS applied to the secondary somatosensory cortex. All groups showed short-term analgesia following interventions without between-group differences. Furthermore, significant pain reduction at 1-month follow-up was found in the secondary somatosensory cortex group.

### 3.2. Patients with SCI

Using a stimulation frequency of 5 Hz, Yılmaz et al. applied HF-rTMS to the primary motor cortex in spinal cord injury (SCI) patients with neuropathic pain—without any superior analgesic effects compared to sham rTMS being observed [19]. By applying 5 days of stimulation with 80% RMT stimulation intensity, Kang et al. evaluated the effects of HF-rTMS of the primary motor cortex in SCI patients with neuropathic pain—with insignificant differences in the numeric rating scales of pain compared with sham rTMS being observed [20]. Sun et al. further discovered that in a similar population, extended stimulation duration using a stimulation intensity of 80% RMT for 6 weeks resulted in a greater reduction of pain than sham rTMS [21].

### 3.3. Phantom Limb Pain

Ahmed et al. applied HF-rTMS to the primary motor cortex in patients with phantom limb pain for five consecutive daily sessions and reported a significant reduction in the VAS of pain at the end of the fifth session (with the effect lasting for 2 months) when compared with the sham group [22]. However, the effect onset was different from that in the study conducted by Malavera et al. [23]. The latter suggested that a significant decrease in pain began on the 15th day after 10 sessions of HF-rTMS, and it could not persist at one month following intervention.

### 3.4. Radiculopathy

Chronic pain induces neuroplastic changes that lead to alterations in pain circuits, rewriting the resting-state cortical activities and maladaptive brain functional network reorganization [24]. Although lumbosacral radiculopathy results from a peripheral lesion, central sensitization and maladaptive neuroplasticity play an important role in its pathophysiology. A study compared the effectiveness of HF-rTMS, anodal transcranial direct current stimulation (tDCS) and sham stimulation in patients with lumbosacral radiculopathy-related neuropathic pain. The authors found superiority of the analgesic effects pertaining to HF-rTMS applied to the primary motor cortex [25]. Repeated daily sessions of rTMS are believed to expand the analgesic effects. Twice-daily sessions or more pulses per session appeared to be tolerable according to the data of previous studies. Moreover, Schulze et al. found that cumulative use of rTMS rather than a single rTMS stimulation had more therapeutic gain [26].

### 3.5. Diabetic Neuropathy

A recent study conducted by Yang et al. found that HF-rTMS applied to the hand primary motor cortex in patients with diabetic peripheral neuropathy offered better analgesic effects 1 week post-treatment compared with sham treatment [27]. Onesti et al. advocated a unique H-coil for HF-rTMS of the leg primary motor cortex that was difficult to focus using the conventional coil [28]. A long-lasting analgesic effect up to 3 weeks post-treatment was documented. Although this deep HF-rTMS technique seemingly provided a longer period of pain reduction, patients suffering from chronic refractory diabetic neuropathic pain might need maintenance treatment thereafter.

### 3.6. Post-Herpetic Neuralgia

In patients with post-herpetic neuralgia, MA et al. [29] and Pei et al. [30] both applied HF-rTMS of the primary motor cortex, with the stimulation frequency of 10 Hz at an intensity of 80% RMT for 10 sessions. They observed significant analgesic effects up to the 3rd post-treatment month. Additionally, Pei et al. compared 10 Hz HF-rTMS, 5 Hz HF-rTMS and sham stimulation in a parallel study design, whereby they demonstrated superior pain reduction and quality-of-life improvement in the 10 Hz HF-rTMS group.

### 3.7. Neuropathic Orofacial Pain

Lindholm et al. proposed the right secondary somatosensory cortex as a novel target for HF-rTMS in patients with refractory neuropathic orofacial pain [31]. A cross-over study was conducted using one session of active HF-rTMS of the right secondary somatosensory cortex, one session of active HF-rTMS of the right sensorimotor cortex and one sham stimulation. The interval between each stimulation was set at 4 weeks, with the sham intervention being interposed between two active rTMS applications. The study results showed a significantly lower numeric rating scale of pain after stimulation of the right secondary somatosensory cortex.

### 3.8. Brachial Plexus Injury

Bonifácio de Assis et al. conducted a randomized controlled trial to evaluate the effects of HF-rTMS and tDCS of the primary motor cortex in patients with neuropathic pain after brachial plexus injury [32]. The active group was provided five daily sessions of HF-rTMS and five daily sessions of tDCS with a washout period of 30 days. Each participant underwent 10 active treatment or sham stimulation sessions in both groups. The results revealed a significant reduction in pain in the treatment group, with the effect lasting for at least one month.

## 4. Parameters/Settings

In our included trials, the frequency of rTMS ranged from 5 to 20 Hz; the intensity ranged between 80% and 120% RMT; and the stimulation pulses ranged from 1000 to 2500 pulses. Until now, there have been no studies using LF-rTMS for managing neuropathic pain. Summarized information from relevant randomized controlled studies, including the characteristics, parameters and outcome, are shown in Table 2.

## 5. Contraindications to rTMS Application

Given that a magnetic field may induce the unsetting or malfunctioning of implanted devices, the absolute contraindication to rTMS is intracranial metallic hardware, such as brain stimulators [37] or electrodes, epidural cortical stimulators, aneurysm clips [38] or coils, stents and programmable ventriculoperitoneal shunts [39]. Since there are no sufficient safety data regarding rTMS for subjects with implanted electronic devices, the contraindication comprises implantations of the cochlear implant, pacemakers, implantable cardioverter defibrillators and spinal cord stimulators. However, a study reported that rTMS seemed to be safe for implanted electronic devices as long as the internal pulse generator was not close to the rTMS coil [40]. Moreover, it is relatively safe for the application of rTMS in patients with newly implanted electronic devices compatible with magnetic resonance imaging, although further large-scaled studies are warranted for long-term follow-up. Furthermore, rTMS is often contraindicated during pregnancy, based on the concerns for fetus safety. However, currently available evidence reported no detrimental effects on the fetus [41]. The summarized contraindications are listed in Table 3.

## 6. Adverse Effects of rTMS

Adverse events, such as scalp discomfort, lightheadedness, headache, nausea, tinnitus and hearing loss, are usually mild to moderate. However, severe ones such as seizure and syncope could occasionally happen. Headache has been reported as the most common adverse event (9.7%), followed by scalp discomfort (9.3%) and nausea (5%) [45,47].

In addition, rTMS produces brief but loud coil click sounds, with the peak sound pressure ranging from 110 to 139 dB for the receivers and from 96 to 125 dB for the operators, exceeding 85 dB, i.e., the threshold that potentially causes transient-to-permanent hearing loss [48]. The click sounds during rTMS could significantly increase the auditory threshold and disturb hearing function after stimulation, even under adequate hearing protection in participants with normal hearing. Nevertheless, the elevation of the auditory threshold did not last more than 1 h after a single rTMS session of 20 min [46].

A study reported that the incidence of rTMS-induced seizure was extremely low and usually self-limited [43]. The estimated risk of seizure per session was 0.05%, 0.03%, 0.06% and 0% in those who received HF-rTMS, LF-rTMS, intermittent theta burst rTMS and continuous theta burst rTMS, respectively [45]. No seizure was reported when rTMS was delivered in accordance with the published guidelines and to individuals without seizure risk [45], e.g., brain tumor, severe head trauma, concussion, neurologic disease or medications that lower the seizure threshold [43]. The summarized adverse events of rTMS are listed in Table 3.

## 7. Mechanism Underlying rTMS in Neuropathic Pain

The analgesic effects of rTMS vary using different parameters. Moreover, the mechanism underlying rTMS is not yet well understood. Therefore, focusing on neuropathic pain, we tried to propose the possible mechanisms regarding HF- and LF-rTMS (Figure 2).

Abbreviations: BDNF, brain-derived neurotrophic factor; DLPFC, dorsolateral prefrontal cortex; HF-rTMS, high-frequency repetitive magnetic stimulation; IL-10, interleukin-10; NMDA, N-methyl-D-aspartate; LF-rTMS, low-frequency repetitive magnetic stimulation; PFC, prefrontal cortex; PMC, premotor cortex; TNF-α, tumor necrosis factor-alpha.

### 7.1. HF-rTMS

The analgesic effect of HF-rTMS of the primary motor cortex was hypothesized to be similar to that of motor cortex stimulation in modulating pain perception and affective–emotional perception. The proposed mechanisms include (1) upregulating the pain threshold via the pathways from the posterior insula and orbitofrontal cortex to the posterior thalamus [49], (2) modulating emotion and mood via pathways from the posterior insula to the caudal anterior cingulate cortex [49] and (3) inhibiting descending pain impulses via pathway from periaqueductal gray to rostroventromedial medulla [50]. The proposed mechanisms resulted in pain reduction of deafferentation pain caused by damage to pain pathways or maladaptive plasticity [51].

In SCI patients, the cortical excitability and analgesic effects of HF-rTMS applied to the motor cortex responsible for hands and legs have been investigated [34]. The use of rTMS reduces pain regardless of the changes in cortical excitability, implying that the analgesic effect might be independent from the excitable status of the motor cortex [34]. In addition, neuro-navigation is a technique designed to localize intracranial structures precisely using a set of brain images [52]. Non-navigated HF-rTMS and neuro-navigated HF-rTMS targeting the primary motor cortex both appeared efficacious, although the latter seemingly produced more extended analgesic effects in patients with focal neuropathic pain [53,54].

HF-rTMS of the primary motor cortex was found to increase the serum beta-endorphin level [22] in patients with phantom limb pain, with a long-lasting analgesic effect. Regarding the role of endogenous opioids in the analgesic effect, HF-rTMS of the primary motor cortex and DLPFC were investigated for pain threshold/intensity followed by naloxone injection. Naloxone is a competitive opioid receptor antagonist and acts as an opioid antidote. Its injection significantly reduced the analgesic effects of the primary motor cortex group, but it did not change the analgesic effects of the DLPFC and sham groups. The findings demonstrated the role of endogenous opioids in rTMS-induced analgesia and suggested different pain-mediated mechanisms in the primary motor cortex and DLPFC territories [55].

Ketamine is known as an N-methyl-d-aspartate (NMDA) receptor antagonist [56]. In another human study, its injection decreased the analgesic effects of HF-rTMS of the primary motor cortex and DLPFC/premotor cortex territories, which was not associated with changes in cortical excitability. This implies that NMDA receptors play a role in the analgesia of rTMS [57]. Interestingly, HF-rTMS was shown to reduce the pain threshold and induce allodynia in rats via NMDA and α-amino-3-hydroxy-5- methyl-4-isoxazolepropionate/kainate receptors [58].

### 7.2. LF-rTMS

LF-rTMS of the prefrontal cortex was found to increase regional brain-derived neurotrophic factor (BDNF), tumor necrosis factor-alpha (TNF-α) and interleukin-10 (IL-10) in rats [59]. BDNF is considered to be an upstream neuro-mediator to induce long-lasting synaptic potentiation [60] and a biomarker of cortical excitability and neuronal activity. It is also known as a nociceptive mediator to amplify ascending pain impulses, leading to hyperalgesia and spinal central sensitization [61]. BDNF may be region-specific; yet, it is downregulated in the hippocampus and upregulated in spinal dorsal horn in rats during noxious stimuli [61]. Higher BDNF levels in the blood and cerebrospinal fluid were found in patients with chronic pain than in the asymptomatic controls [62]. Finally, IL-10 serves as an anti-inflammatory mediator. This evidence implies that neurotrophin and anti-neuroinflammatory cytokines might play a role in the analgesic effects of LF-rTMS.

## 8. rTMS for Comorbidities of Neuropathic Pain

Patients suffering from chronic neuropathic pain often have comorbidities, such as depression and mood disorders. The prevalence rates of depression and major depressive disorders among patients with chronic neuropathic pain were 65.6% [63] and 16.5% [64], respectively.

Imbalanced connectivity of the DLPFC was found in patients with depression [65]. HF-rTMS was applied on the left DLPFC to facilitate neuronal activity and to treat depression [66]. The DLPFC is activated in response to noxious stimuli and is associated with pain suppression through the descending pain pathways [67]. Moreover, the activity of the left DLPFC was negatively correlated with pain [68]. A neurophysiological study revealed that conditioned transcranial magnetic stimulation of the frontal cortex modulated the parameters of motor-evoked potentials, suggesting the existence of cortico-cortical connection [69].

A meta-analysis demonstrated that HF-rTMS of the DLPFC induced a short-term analgesic effect [70], while HF-rTMS of the primary motor cortex showed more evident analgesic effects in patients with neuropathic pain [71]. Moreover, a randomized controlled trial concluded that the analgesic effects of HF-rTMS were not associated with the indirect improvement of depression and were not mediated/predicted by comorbid psychiatric disorders [72].

## 9. Discussion/Future Perspective

The complexity of neuropathic pain and mechanisms behind rTMS are not completely understood. A recent meta-analysis found that neuropathic pain related to SCI, diabetic neuropathy and post-herpetic neuralgia appeared to be effective after HF-rTMS [71], whereas LF-rTMS showed insignificant analgesic effects. However, there is still lack of evidence for applying rTMS in the general population due to heterogeneity and the small number of available clinical studies.

In combination with our included trials and aforementioned studies, 10 Hz HF-rTMS of the primary motor cortex is recommended as the initial setting for the treatment of neuropathic pain. Regarding stimulation intensity, pulses and sessions, identical parameters have not been determined yet. Similarly, a practical algorithm for applying rTMS for chronic pain was proposed by Lefaucheur et al. in 2019 [73]. Clinically relevant evidence did not support somatotopic effects of HF-rTMS on neuropathic pain [74]. HF-rTMS of the primary motor cortex corresponding to the painful site was less effective than that of the area next to the cortical representation of the painful zone [75]. Hence, 10 Hz HF-rTMS at the intensity of 80–90% RMT could be started by stimulating the primary motor cortex of hands in all patients regardless of the pain location/origin. The stimulation side of the primary motor cortex of hands could be contralateral to the painful side for lateralized neuropathic pain or the left brain cortex for diffuse neuropathic pain. After six–seven sessions, the physicians should re-evaluate whether pain reduction could reach more than 30% of the baseline assessment or two points on a 0–10 numeric rating scale. If the patient does not respond well to the initial session, an alternative stimulating target would be the primary motor cortex corresponding to the sensory/motor processing of face and legs. If there is still no response, the last, but not least, target should be shifted to the left DLPFC, using the beam F3 localization method [76] at 100–110% RMT. The secondary somatosensory cortex is located closely to the insular cortex, which is known to have an important role in pain perception. Strong functional connections between the secondary somatosensory cortex and insular cortex had been found during painful stimuli [77]. rTMS induces neuroplastic changes not only at the stimulation site but also in distant locations probably through cortico–cortical connections and subcortical networks. Therefore, rTMS may exert its analgesic effect in multiple locations of neuronal networks. In cases with good responses to the initial protocol, HF-rTMS can be gradually titrated down from twice a week to once per month [78].

Nevertheless, there are several limitations to the present narrative review. Firstly, despite the fact that we defined inclusion and exclusion criteria and systematically conducted a literature search to reduce selection bias, some studies may have been inevitably omitted. Given the small number of included studies in each etiology category, we had difficulty in pointing out solid and strong opinions without too much bias in only one–two studies. Therefore, we only described their study designs and presented the results objectively in Table 1. Secondly, the algorithm of rTMS application was not tailored according to different diseases. Meanwhile, the conclusion regarding the effective parameters of rTMS from existing meta-analyses might be overgeneralized and not be suitable for various causes of neuropathic pain.

Indeed, the duration of the analgesic effects of HF-rTMS varies from hours [34] to days [78] or one month [31] across different diseases and stimulation parameters. A standard protocol, e.g., a minimal number of sessions in the induction phase, and duration of the maintenance phase and interval between sessions, has not been established. As such, future high-quality studies are definitely awaited.

## 10. Conclusions

rTMS is a potentially effective and safe treatment of neuropathic pain. Current evidence supports the use of 10 Hz HF-rTMS of the primary motor cortex to reduce neuropathic pain, especially in patients with SCI, diabetic neuropathy and post-herpetic neuralgia. However, the lack of standardized protocols impedes the universal use of rTMS in neuropathic pain. rTMS is hypothesized to achieve analgesic effects by upregulating the pain threshold, inhibiting pain impulse, modulating the brain cortex, altering imbalanced functional connectivity, regulating neurotrophin and increasing endogenous opioids and anti-inflammatory cytokines. Further studies are warranted to explore the differences in the parameters/settings of rTMS for neuropathic pain caused by different diseases.

## Figures and Tables

**Figure 1 life-13-00258-f001:**
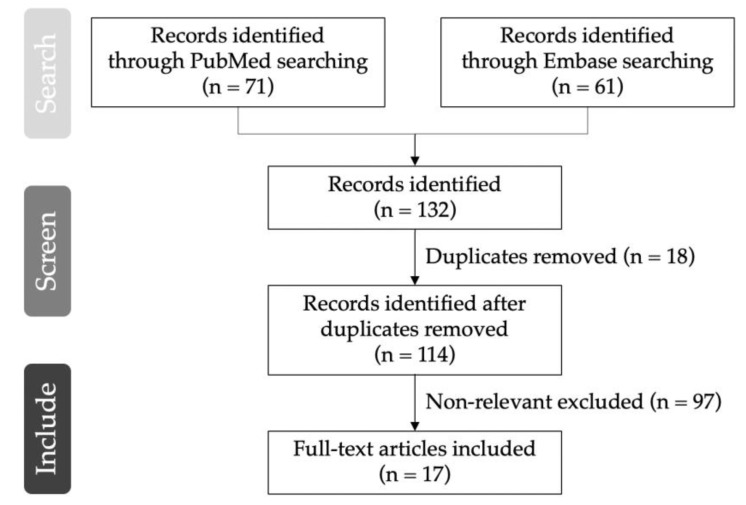
Flow diagram for the literature search.

**Figure 2 life-13-00258-f002:**
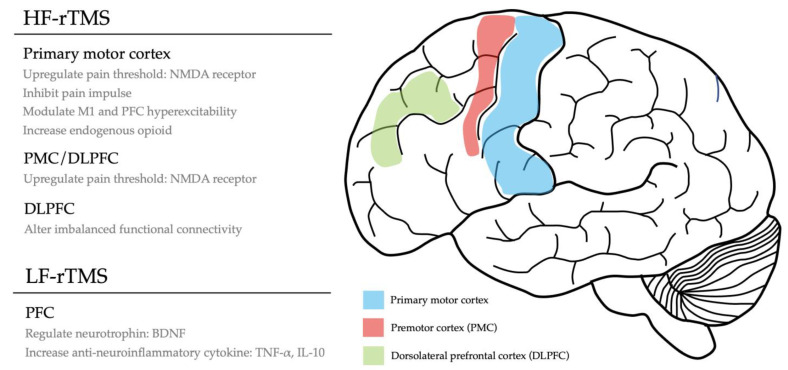
Schematic diagram pertaining to the underlying mechanism of repetitive transcranial magnetic stimulation in alleviating neuropathic pain.

**Table 1 life-13-00258-t001:** Indications of rTMS in neuropathic pain.

◆ Central neuropathic pain: ❿ Post-stroke; ❿ Spinal cord injury; ❿ Phantom limb pain. ◆ Peripheral neuropathic pain: ❿ Radiculopathy; ❿ Diabetic neuropathy; ❿ Post-herpetic neuralgia; ❿ Neuropathic orofacial pain (e.g., trigeminal neuralgia); ❿ Brachial plexus injury.

**Table 2 life-13-00258-t002:** Characteristics, parameters and results of included randomized controlled studies.

Author (Year)	Disease	Baseline Characteristics: Mean Age/Number (Male/Female)	rTMS (*n* = xx), Control or Sham (*n* = xx)	rTMS Site	rTMS Frequency	rTMS Intensity	rTMS Pulses	rTMS Session	Outcome
de Oliveira et al. (2014) [18]	Post-stroke	56.3 (l1/10)	11, 10	Left PMC/DLPFC	10 Hz	120% RMT	1250	Daily with 2-day weekend interval for a total of 10 consecutive sessions	No differences in pain reduction over 1 month compared to the hand motor cortex group and sham group
Ojala rt al (2022) [33]	Post-stroke	55.8 (8/9)	17, 17	S2 contralateral to painful site	10 Hz	90% RMT	5050 (train duration, 10 s; intertrain pause, 50 s)	10 sessions	Significant reduction of weekly pain intensity in S3 group compared with the sham group
Kang et al. (2009) [20]	SCI	54.8 (6/5)	11, 11	FDI motor cortex	10 Hz	80% RMT	1000 (train duration, 5 s; intertrain pause, 55 s)	Daily, 5 consecutive sessions	No differences in average NRS reduction compared to the sham group
Jetté et al. (2013) [34]	SCI	50 (11/5)	16, 16	Hand: FDI motor cortex Leg: vertex motor cortex	10 Hz	Hand: 90%RMT Leg: 110%RMT	2000 (train duration, 5 s; intertrain pause, 25 s)	One session	About 10% NRS reduction over the first 49 h compared with the sham group
Yılmaz et al. (2014) [19]	SCI	38.6 (16/0)	9, 7	Vertex motor cortex	5 Hz	110% RMT	1500 (train duration, 5 s; intertrain pause, 25 s)	Daily for a total of 10 consecutive sessions	Not superior to the sham group
Nardone et al. (2017) [35]	SCI	43.1 (9/3)	6, 6	PFC/DLPFC: 6 cm anterior to the FDI motor cortex	10 Hz	120% RMT	1250 (train duration, 5 s; intertrain pause, 25 s)	5 times per week for 2 weeks for a total of 10 consecutive sessions	Significant VAS reduction over 1 month compared with the sham group
Sun et al. (2019) [21]	SCI	37 (15/2)	11, 6	Hand motor cortex	10 Hz	80% RMT	1200 (train duration, 1.2 s; intertrain pause, 3 s)	Daily with 1-day interval per week for a total of 6 weeks	Greater NRS reduction after 2 weeks of rTMS sessions than the sham group
Zhao et al. (2020) [36]	SCI	41.6 (NA)	24, 24	Hand motor cortex	10 Hz	90% RMT	1500 (intertrain pause, 3 s)	Daily with 1-day interval per week for a total of 3 weeks	Significant NRS reduction on the 3rd day and 1st week post-rTMS compared with the sham group
Ahmed et al. (2011) [22]	Phantom limb pain	52.0 (13/140	27	Motor cortex corresponding to the stump of painful site	20 Hz	80% RMT	NA (train duration, 10 s)	Daily, 5 consecutive sessions	Significant VAS reduction over 2 months compared with the sham group; 55%, 52% and 39% VAS reduction on the day, 1st month and 2nd month post-rTMS, respectively
Malavera et al. (2016) [23]	Phantom limb pain	33.9 (50/4)	27, 27	Contralateral leg motor cortex	10 Hz	90% RMT	1200 (train duration, 6 s; intertrain pause, 54 s)	Daily, 10 consecutive sessions	30.44% greater mean VAS reduction on the 15th day post-rTMS, and no differences on the 30th day post-rTMS, than the sham group
Attal et al. (2016) [25]	Lumbosacral radiculopathy	52.7 (17/18)	21, 11	Thenar motor cortex	10 Hz	80% RMT	3000 (train duration, 10 s; intertrain pause, 20 s)	Daily, 3 consecutive sessions	30.4% mean pain reduction
Onesti et al. (2013) [28]	Diabetic neuropathy	70.6 (14/9)	23, 23	Leg motor cortex	20 Hz	100% RMT	1500 (intertrain pause, 30 s)	Daily, 5 sessions	Significant VAS reduction over 3 weeks compared with the sham group
Yang et al. (2022) [27]	Diabetic peripheral neuropathy	60.4 (11/9)	10, 10	Left APB motor cortex	10 Hz	90% RMT	1000 (train duration, 5 s; intertrain pause, 55 s)	Daily, 5 sessions	Significant NRS reduction from 6.5 ± 0.9 to 3.6 ± 0.7 1 day post-rTMS; non-significant NRS reduction 1 week post-rTMS compared to the sham group
Ma et al. (2015) [29]	PHN	66.4 (20/20)	20, 20	Motor cortex corresponding to a painful site	10 Hz	80% RMT	1500 (train duration, 5 s; intertrain pause, 3 s)	5 times per week for 2 weeks for a total of 10 consecutive sessions	16.9% mean VAS reduction over 6 months
Pei et al. (2019) [30]	PHN	66.2 (30/30)	20, 20, 20	Motor cortex corresponding to a painful site	10 Hz	80% RMT	10 Hz group: 1500 (train duration, 0.5 s; intertrain pause, 3 s)	Daily, 10 sessions	Superior VAS reduction in 10 Hz group over 3 months compared with 5 Hz and sham groups
Lindholm et al. (2015) [31]	Drug-resistant neuropathic orofacial pain	57.7 (7/13)	10, 6	Right S2	10 Hz	90% RMT	2500 (train duration, 5 s; intertrain pause, 15 s; 15 min break in the middle of session)	One session	Lower NRS of pain over 1 month than S1/M1 or sham groups
Bonifácio de Assis et al. (2022) [32]	Traumatic BPI	32.8 (20/0)	12, 8	Contralateral hand motor cortex	10 Hz	90% RMT	2500 (train duration, 10 s; intertrain pause, 17 s)	Daily, 5 consecutive days	Superior continuous pain, paroxysmal pain and anxiety reduction over 1 month compared with the sham group

Abbreviations: APB, abductor pollicis brevis muscle; BPI, brachial plexus injury; FDI, first dorsal interosseous muscle; NA, not available; NRS, numeric rating scale; PHN, post-herpetic neuralgia; PMC/DLPFC, premotor cortex/dorsolateral prefrontal cortex; rTMS, repetitive transcranial magnetic stimulation; RMT, resting motor threshold; S1/M1, sensorimotor cortex; S2, secondary somatosensory cortex; SCI, spinal cord injury; VAS, visual analog scale.

**Table 3 life-13-00258-t003:** Contraindications to and adverse effects of rTMS.

Contraindications to rTMS	Adverse Effects of rTMS
Absolute Intracranial metallic hardware: brain stimulators [37] or electrodes, epidural cortical stimulators, aneurysm clips [38] or coils, stents and programmable ventriculoperitoneal shunts [39] Relative Implanted electronic device: cochlear implants [40], vagus nerve stimulators [42], pacemakers or implantable cardioverter defibrillators and spinal cord stimulators History of seizure [43], epilepsy and transcranial magnetic stimulation -related seizure History of syncope History of vascular, traumatic, tumoral, infectious, or metabolic brain lesion [44] Drugs that potentially lower seizure threshold Special consideration Pregnancy: currently available evidence reported no detrimental effects on the fetus [41]	Seizure (<1%) [43,45] Syncope Scalp discomfort, burn and pain Lightheadedness Headache (most common) Nausea and vomiting Tinnitus and hearing loss [46]

## Data Availability

Data are contained within the main text of the manuscript.

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
