# Peer review of "Application of Repetitive Transcranial Magnetic Stimulation in Neuropathic Pain: A Narrative Review"

_life, 2023, doi:10.3390/life13020258_

Round 1
Reviewer 1 Report
This manuscript was to provide an up-to-date over-view of rTMS in treating neuropathic pain and summarize the treatment protocols and related adverse effects from existing clinical trials.This filed is interesting.However, the revised version will have a better impact in the area and the readers of the journal will have a better understanding of this manuscript.
1.The manuscript still needs some minor copyediting for grammatical errors, typos, and word omissions.
2. In the section of Introduction, the differences between this review and previous reviews are not clearly introduced.
3.In the section of “3. Therapeutic application of rTMS in neuropathic pain”, more related studies need to cited and discussed.
4. The review article should not simply list the research results, but should summarize the results of previous research, and come to their own views and opinions from authors. The authors do not make their own points in most of the sections in the present manuscript.
5.The most suitable stimulating sites and parameters for different diseases can be discussed in the section of Discussion.
Author Response
Dear reviewer 1:
Thank you for your kind comments for our manuscript. Our responses to your queries have been detailed in the rebuttal letter. Again, we appreciate your efforts in improving the quality of the present manuscript.

Reviewer 2 Report
The authors have reviewed the protocols and adverse effects in the application of rtms in patients with neuropathic pain.
Authors should report the inclusion and exclusion criteria of the studies on which the review is based. They should also indicate the number of studies found and discarded.
The authors focus their review on 2 studies for the different pathologies (except Sci), so it cannot be said that there is scientific evidence nor can the protocol be given as a standard for these pathologies.
Figure 1 is not representative of the problem the authors want to explain, the figure should be changed/modified and better explained in the legend.
Author Response
Dear reviewer 2:
Thank you for your kind comments for our manuscript. Our responses to your queries have been detailed in the rebuttal letter. Again, we appreciate your efforts in improving the quality of the present manuscript.

Reviewer 3 Report
Thank you for giving me the opportunity to review this manuscript.
This article was an up-to-date review of the effects of rTMS on neuropathic pain.
I think it is necessary to revise the manuscript.
Please focus on how to optimize study protocols, as the authors mentioned in the introduction.
1) Please describe briefly the inclusion criteria and exclusion criteria of this review. Were included studies all randomized controlled trials?
2) If possible, please specify the methods and results to assess risk of bias in the included studies. If it is impossible, please describe it in the limitations.
3) Please present baseline characteristics of included studies.
4) In studies by de Oliveira et al [17], rTMS was applied over the premotor cortex/dorsolateral prefrontal cortex (DLPFC), and the number of pulses were 1200, and sample size was around 10 in each arm. I think that the target area (not premotor area only), the number of pulses, the number of sessions, the infarct area, and sample size probably affected the outcomes. What do you think of that? What kind of further studies are warranted to optimize the rTMS protocol in patients with central post stroke pain? Was the duration of its effect clinically meaningful?
5) Sun et al [20] found that the number of sessions may affect the effects of rTMS on spinal cord injury, but please explain what kind of other factors (sample size and number of pulses) may affect the outcomes? Was the duration of its effect clinically meaningful?
6) Please delete the sentences of tDCS studies to focus on rTMS in this review. For example, please delete the sentences of " However, the effectiveness of tDCS was similar to sham stimulation.". Furthermore, please delete the sentence of "Moreover, the health-related quality of life assessment in physical and mental aspects improved in the group receiving active rTMS. " to focus on the effects on pain.
7) Please explain why rTMS was effective in patients with Lumbosacral radiculopathy. Was the number of sessions (only three times) not important in patients with Lumbosacral radiculopathy? Did the number of pulses affect its effects? Did the Lumbosacral radiculopathy in itself affect the outcome? Was the duration of its effect clinically meaningful?
8) In the study by Yang et al, you described that "Average NRS reduction from 6.5 ± 0.9 to 3.6 ± 0.7 at 1 day post-rTMS and to 5.3 ± 1.1 at 1 week post-rTMS", but does it mean that post rTMS scores were better that pre-rTMS scores? What about the between group differences of mean changes from baseline? I think this sentence did not mean that active rTMS was effective than sham rTMS. Why rTMS was applied over the left primary motor cortex of abductor pollicis brevis muscle in patients with Diabetic peripheral neuropathy? Why was this protocol effective in spite that the number of sessions and pulses were small.
9) In Table 2, the authors showed in the study by Pei et al that “10 Hz group: 1500 (train duration: 0.5 sec, intertrain pause: 3 sec)”, but what does this protocol mean? What about the non-10 Hz group? Was that parallel-group RCT? Please delete the words of “QoL and sleep quality” to focus on pain. Please describe how much “superior VAS reduction in 10 Hz group over 3 months compared to 5 Hz and sham groups” was found in patients with Post-herpetic Neuralgia.
10) In Table 2, the authors described in the study by Lindholm et al that rTMS over right secondary somatosensory cortex improved numeric rating scale, but what did the numeric rating scale mean and how much rTMS improved this scale. What did the result mean? One-session rTMS over right secondary somatosensory cortex was extremely effective in alleviating pain in patients with Drug-resistant neuropathic orofacial pain? Why rTMS was placed over the secondary somatosensory cortex in patients with Drug-resistant neuropathic orofacial pain? Where were patients painful?
11) In the studies by Bonifácio de Assis et al, you described that “All participants were randomized in 1-to-1 ratio to the active or sham group.”. If all of the design of studies was randomized controlled trial in this review, please delete the sentence because it is not meaningful.
12) What does the sentence of “The former was provided 5 daily sessions of HF-rTMS and 5 143 daily sessions of tDCS with a washout period of 30 days.” mean? Patients in the active group received both rTMS and tDCS? Please focus on rTMS studies only in this review. Please delete the sentences of anxiety to avoid misleading. How much rTMS improved continuous pain, paroxysmal pain? Why was rTMS effective in patients with Traumatic brachial plexus injury? Why rTMS was placed over Contralateral hand motor cortex in patients with traumatic brachial plexus injury?
13) In the mechanism section, I cannot understand that why high-frequency rTMS modulate M1 and PFC cortical excitability in spite that high frequency rTMS usually enhances cortical excitability. Please explain it.
14) Please delete the sentences of “On the contrary, a functional magnetic resonance imaging study in the rat model with neuropathic pain revealed that motor cortex stimulation attenuated the hyperexcitability of primary somatosensory cortex and prefrontal cortex and, therefore, alleviated pain intensity.”, because you must focus on human studies to understand the mechanisms. Furthermore, please delete the sentences of “Similarly, using functional near-infrared spectroscopy, HF-rTMS over the primary motor cortex was found to suppress its hyperexcitability as well as that of the premotor cortex. Neuropathic pain could therefore be ameliorated in SCI patients.”, because the effect of functional near-infrared spectroscopy on the detection of cortical excitability of whole brain areas was not established at all.
15) You described that “Imbalanced connectivity of DLPFC was found in patients with depression. HF-rTMS has been applied on left DLPFC to facilitate neuronal activity and to treat depression. DLPFC is activated in response to noxious stimuli and is associated with pain suppression through the descending pain pathways”, while you explained “HF-rTMS over the primary motor cortex showed more evident analgesic effects in patients with neuropathic pain [65]. Moreover, a randomized conrolled trial concluded that the analgesic effects of HF-rTMS was not associated with indirect improvement of depression and was not mediated/predicted by comorbid psychiatric disorders” I think this does not make sense. DLPFC should note be the target of rTMS in alleviating pain only. Please delete the sentences of “HF-rTMS over the primary motor cortex might activate DLPFC through its functional connectivity. Hence, HF-rTMS has been increasingly applied on left DLPFC as an alternative target to alleviate chronic neuropathic pain that may coexist with depressive disorders.” to focus on pain and its mechanisms only. I think that rTMS should not be used as an indirect effect only in clinical practice.
16) Please delete the following sentences and add different discussions.
". After 6 to 7 sessions, the physicians should re-evaluate whether pain reduction could reach more than 30% of the baseline assessment or two points on a 0-10 numeric rating scale. If the patient does not respond well to the initial session, an alternative stimulating target would be the primary motor cortex corresponding to the sensory/motor processing of face and legs. If there is still no response, the last target should be shifted to the left DLPFC, using a beam F3 localization method [70] with 100-110% RMT. In cases with good responses to the initial protocol, HF-rTMS can be gradually titrated down from twice a week to once per month"
That is because how the changing the target site was effective was not shown at all in previous studies, and because not only the target site but also the type of disease, the number of sessions, the number of pulses, the sample size, and the study outcomes may affect the pain. Please discuss those factors as well in the discussion sections.
It is mandatory to revise the manuscript substantially.
Author Response
Dear reviewer 3:
Thank you for your kind comments for our manuscript. Our responses to your queries have been detailed in the rebuttal letter. Again, we appreciate your efforts in improving the quality of the present manuscript.

Round 2
Reviewer 2 Report
The authors have responded well to the comments, providing the necessary data for this narrative review and thus increasing readers' interest in it. I have no new comments.
Reviewer 3 Report
Thank you for revising the manuscript.
I think this manuscript would be suitable for publication.